# *DICER1* Mutational Spectrum in Intracranial CNS-Neoplasias—A Review and a Report from the CNS-InterREST GPOH Study Center

**DOI:** 10.3390/cancers17091513

**Published:** 2025-04-30

**Authors:** Selma Manea, Victoria E. Fincke, Michael C. Frühwald, Dominik Sturm, Barbara von Zezschwitz, Pascal D. Johann, Marlena Mucha

**Affiliations:** 1Pediatrics and Adolescent Medicine, Swabian Children’s Cancer Center, University Hospital Augsburg, 86156 Augsburg, Germany; 2Bavarian Cancer Research Center (BZKF), 85156 Augsburg, Germany; 3Department of Pediatric Oncology, Hematology, & Immunology, Heidelberg University Hospital, Division of Pediatric Glioma Research, German Cancer Research Center (DKFZ) and Hopp Children’s Cancer Center Heidelberg (KiTZ), 69120 Heidelberg, Germany; 4Klinik für Pädiatrie m. S. Onkologie/Hämatologie, Charité-Universitätsmedizin Berlin, 10117 Berlin, Germany

**Keywords:** pediatric cancer, *DICER1*, tumor predisposition syndrome, DICER1 syndrome

## Abstract

Understanding how *DICER1* gene mutations are linked to rare childhood brain tumors is challenging, mainly because these mutations are uncommon. This study reviewed existing research and examined data from a German pediatric tumor registry to explore how often *DICER1* mutations appear in tumors like ETMR and intracranial sarcomas. We found that these mutations are sporadic, but when they occur, they often affect a specific part of the gene that may interfere with how certain RNA molecules are processed—possibly helping tumors form. Although *DICER1* is usually seen as a tumor-suppressing gene, some findings suggest it could also act in ways that promote cancer, depending on the mutation. These results highlight the need for more research into how different *DICER1* mutations work. This information could support earlier diagnosis for families who already know they carry a *DICER1* mutation.

## 1. Introduction

The plethora of clinical manifestations of DICER1 syndrome is in contrast to the limited biological understanding of the function of the aberrant DICER1 protein. The factors influencing penetrance in DICER1 syndrome, which determines whether carriers develop tumors, remain unknown. For instance, even among individuals with a confirmed DICER1 predisposition, the incidence of tumors is relatively low (ranging between 20% and 30%) [1]. Furthermore, the distribution and relative contributions of germline versus somatic *DICER1* mutations in intracranial sarcomas and ETMRs remain poorly understood. This concise review provides an overview of the biological role of *DICER1*, the most important intracranial manifestations of DICER1 syndrome, and reviews somatic and germline distributions in four prototypic DICER1-associated tumors (intracranial sarcoma, pineoblastoma, ETMR, and pleuropulmonary blastoma).

## 2. Biological Function and Structure of DICER1

The *DICER1* gene is located on chromosome 14q32.13 and encodes the DICER1 endoribonuclease protein, which plays a crucial role in RNA processing and RNA interference pathways [2]. In brief, DICER1 cleaves dsRNA and pre-miRNA into mature siRNA and miRNA, which are then incorporated into the RNA-induced silencing complex (RISC) and guided to target messenger RNAs (mRNAs) with complementary sequences. This leads to the degradation of target mRNAs or inhibition of their translation, effectively silencing specific genes [3]. Through its involvement in post-transcriptional regulation, DICER1 influences various biological processes, including embryogenesis and cellular differentiation, and is therefore frequently deregulated in cancer.

The structure of the DICER1 protein has recently been resolved using Cryo-EM [4], and it resembles the letter “L” (Figure 1). The PAZ and Platform domains located at the top of the “L” are involved in dsRNA binding. Specifically, the PAZ domain is essential for binding the 3’-overhang of the dsRNA, while the Platform domain contains a binding pocket for the 5’-phosphate of the dsRNA [5]. Toward the middle of the “L” structure is the catalytic core of the enzyme, comprising two RNAse domains, RNAse IIIa and RNAse IIIb, which are responsible for dimerization and dsRNA cleavage. The product of each cleavage is 3p or 5p miRNA from RNAse IIIa and RNAse IIIb, respectively. The bottom of the “L” contains the Helicase domain, which binds to pre-miRNAs and the dsRNA-binding fold [3,6]. Two different double-stranded RNA-binding protein-binding sites can also be found in the same region: trans-active response RNA-binding protein (TRBP) and protein activator of PKR (PACT) [7].

With the ability to cleave, DICER1 plays an important part in the RNAi pathway. It can process pre-miRNA (miRNA precursor hairpins) into miRNA by binding to PACT and long dsRNA into siRNA by binding to TRBP. After processing, DICER1 loads small RNA onto Argonaute proteins (AGO) to initiate the RNA-induced silencing complex (RISC); therefore, it is part of the RNA-induced silencing complex loading complex (RLC) [8].

Additionally, DICER1 has been shown to mediate R-loop processing by cleaving RNA within DNA-RNA hybrids [9]. As increased accumulation of R-loops may lead to genome instability [10], DICER1 plays an important role in maintaining non-harmful levels in cells. Given the universal role of DICER1 in the processing of miRNAs and its involvement in R-loop regulation, the broad clinical impact of *DICER1* germline mutations and the various manifestations of *DICER1* mutations are not surprising.

## 3. Epidemiology of DICER1 Syndrome

Several large cohort studies have examined the prevalence of *DICER1* variants in the population. Mirshahi et al. studied whole exome sequencing data of 92,296 participants, of whom 25 individuals displayed a loss-of-function variant in *DICER1* (12 different LOFVs were found) [1]. The overall frequency of LOFV in the whole cohort was 1/4600, while other studies have estimated its frequency to be lower at 1/10,600 [2].

The penetrance of DICER1 syndrome remains incompletely understood and appears to vary depending on the specific tumor type; overall, the lifetime risk of developing any tumor has been estimated to be between 25% and 30% [1]. The fraction of de novo mutations is roughly 20%, with most cases occurring in the context of index families [11], reflecting a strong familial inheritance pattern. In a recent longitudinal study by Stewart et al., 207 carriers of *DICER1* pathogenic variants were tracked for their risk of developing any neoplasia up to a certain age: Only 5.3% of the patients developed a tumor before the age of 10 years and of 31.5% before the age of 60 [12]. While a systematic review of all tumor types associated with this condition is beyond the scope of this review, we focus on the most pertinent entities that occur during (early) childhood and affect the central nervous system (CNS).

## 4. *DICER1*—Mutations and Associated Cancer Types

As outlined above, DICER1 plays a crucial role in regulating RNA processing, the maturation of small non-coding RNAs, and RNA-mediated gene silencing; therefore, it is not unexpected that the *DICER1* gene is often mutated in a variety of cancers [13]. Only one altered allele can cause DICER1 syndrome, which manifests as a predisposition to developing pleuropulmonary blastomas (PPBs), pulmonary cysts, or ovarian tumors [11]. DICER1 syndrome, which mainly affects children and young adults, is usually caused by a germline loss-of-function *DICER1* mutation. However, the development of the disease requires the acquisition of a second somatic mutation in the other allele. The secondary alteration predominantly localizes to the genomic region encoding RNase III domains [2,14]. Studies suggest that *DICER1* mutations in different domains may play distinct roles in the oncogenesis of various primary CNS tumors [13]. This substantiates the notion that *DICER1* is a tumor suppressor gene that aligns with the two-hit hypothesis of tumorigenesis [3].

Mutations affect the amino acid S1344 within the RNase IIIa domain and amino acids E1705, D1709, and E1813 within RNase IIIb [15]. According to a functional model proposed by Lambo et al., while a certain degree of functionality is preserved in DICER1, RNAse IIIb domain mutations lead to a reduced abundance of 5p dominant miRNAs—leaving the 3p processing pathway uncompromized [16]. The imbalance between 3p and 5p miRNAs leads to significant changes in gene expression profiles, thus contributing to tumorigenesis. Additionally, the inability of mutated DICER1 to properly cleave RNAs leads to the accumulation of harmful levels of R-loops [16].

Regarding the effect of *DICER1* mutations on central nervous system development, they have been shown to disrupt neural crest differentiation, leading to midbrain and cerebellum malformations, dopaminergic neuron defects, and cortical and hippocampal structural abnormalities due to decreased microRNA levels [17]. Additionally, macrocephaly has been observed in individuals with *DICER1* mutations [18]. Notably, this feature was more prevalent in patients with mosaic *DICER1* RNase IIIb mutations. In a study analyzing growth data from 67 *DICER1* mutation carriers, macrocephaly and symmetric overgrowth were reported in some but not all patients with these mosaic mutations [18].

In the sections below, we outline the clinical and genetic backgrounds of three intracranial DICER1-associated cancer types in pediatrics: ETMR, sarcomas, and pineoblastomas. We also include a non-intracranial pediatric entity, pleuropulmonary blastoma.

## 5. Embryonal Tumor with Multilayered Rosettes (ETMR)

ETMRs are aggressive, rapidly growing brain tumors (supra- or infratentorial) that mainly occur in young infants [15]. Prior to establishing ETMR as a separate diagnostic group, these tumors have been described by their histological appearance as embryonal tumors with abundant neuropil and true rosettes (ETANTR), ependymoblastomas (EBL), or medulloepitheliomas (MEPL) [19]. Only upon the discovery of the amplification of 19q13.42, containing the C19MC miRNA cluster, were the three histological types unified into one entity, with C19MC amplification being the diagnostic marker. Additionally, LIN28A may serve as a surrogate marker for ETMR in immunohistochemistry. However, it has been shown that not all LIN28A-positive ETMRs have the C19MC amplification [15]. This may be attributed to the limited specificity of LIN28A as a marker. However, methylation profiling has revealed that 10% of all cases clustering with ETMR lack amplification of the C19MC locus, suggesting the presence of alternative molecular drivers in the pathogenesis of ETMR.

Many of these, along with 5% of all ETMRs, display *DICER1* alterations. In a hallmark publication from Lambo et al., these mutations were shown to affect mainly the residues within the RNase IIIb domain, which leads to increased loading of the 3p arm on AGO because the 5p arm of miRNA is not spliced accurately and, therefore, is degraded more often [16]. This is a more substantial challenge for miRNA clusters that require 5p for loading than for those that load the 3p arm. Although the biological explanation for this process is conclusive, it is unclear whether the more common mechanism of ETMR formation, C19 miRNA amplification, and *DICER1* mutations converge on the same pathways. The occurrence of *DICER1* mutations in ETMR is exceedingly rare; in 31 cases registered in the CNS-InterREST GPOH database, we identified only one with a *DICER1* mutation.

Given the development of *DICER1* mutant ETMRs in very young infants (Table 1), they may represent the earliest-onset tumor type under the DICER1 syndrome umbrella. As such, a common challenge is the inclusion and position of radiotherapy in the treatment regimens. The ESCP guidelines emphasize the importance of early radiotherapy and affirm that there is no standard care chemotherapy in these patients. Importantly, genetic counseling is necessary and should also include considerations regarding the patients’ parents.

## 6. Intracranial Sarcoma

Intracranial Sarcomas represent a rare tumor group that potentially develops from multipotent mesenchymal cells within the meninges [20]. In the literature, this entity is referred to as Primary Intracranial Sarcoma with *DICER1* alteration (PIS DICER) [21] or primary DICER1-associated central nervous system sarcoma (DCS) [2]. It primarily occurs in young adults and is highly malignant [22]. In primary intracranial sarcomas, *DICER1* mutations are observed in a significant proportion of cases, with studies reporting mutations in up to 93% of these tumors [23]. DICER1 sarcomas are often accompanied by *TP53* mutations and an altered MAP kinase signaling pathway [24]. Clinically, no established therapy concept exists; in many cases, a combined approach using radiation and chemotherapy is used, although there is no conclusive data regarding the superiority of any treatment regimen. Two case studies from Peru and Colombia mainly used two to three cycles of chemotherapy followed by radiation. In South American studies by Cardona et al., ICE (Ifosfamide, Carboplatinum, and Etoposide)-based regimens achieved a median survival of 30.8 months, while the Peruvian cohort seemed to display a markedly better survival (8 years vs. 20 years) [2,25]. Whether there is a genotype−phenotype correlation for these intracranial sarcomas remains unresolved.

## 7. Pineoblastoma (PinB)

Pineoblastomas (PinB) are rare primitive neuroectodermal tumors that arise in the pineal gland [26]. *DICER1* mutations have been identified in approximately 26% to 50% of pediatric PinB cases [23]. Unlike the tumor-specific somatic RNase III hotspot mutation, loss of heterozygosity in the wildtype allele is more common in DICER1-related pineoblastoma than in other entities [14]. Clinically, pineoblastoma is an aggressive brain tumor with low progression-free and overall survival rates. Abdelbaki et al. reported on the experience with PinB in the Headstart trials: from 23 patients (all below six years of age as per the inclusion criterion of the study), only three survived longer than 5 years. Although the number of patients included in this study was low, patients who received radiation therapy displayed better survival. Notably, the HS concept includes craniospinal radiation but not local radiation [27].

In a recent landscaping paper comprising 195 cases of pineal region tumors, Pfaff et al. proposed a classification of these tumors comprising five entities [28]. Among them are three pineoblastoma subtypes. Subtype PB-Grp1B displayed the highest rate of *DICER1* mutations (75%), and Group 1A (defined primordially by its methylation pattern) displayed aberrations in either *DICER1* or *DROSHA*.

Overall, pineoblastoma is an aggressive embryonal brain tumor that typically necessitates a combination of chemotherapy and, when clinically appropriate, radiation therapy to achieve therapeutic efficacy. A prospective evaluation of the molecular subgroups is necessary to determine whether subtle survival differences become more pronounced in larger patient groups.

## 8. Pleuropulmonary Blastoma (PPB)

Although this tumor is rare in absolute numbers, it is perceived to be an important marker for DICER1 syndrome, as approximately 70% of children with pleuropulmonary blastoma (PPB) carry germline *DICER1* mutations [29,30]. PPB is classified into three types: PPB I, characterized by a cystic lesion; PPB II, containing both cystic and solid areas; and PPB III, consisting of solid regions only [29,31]. Over 70% of PPB cases have a germline loss-of-function mutation and a second somatic RNase IIIb domain mutation. These hotspot mutations primarily affect residues E1705, D1709, G18009, D1810, and E1813 [32]. In our review, we used pleuropulmonary blastoma as a reference for a non-CNS tumor with a well-characterized mutational distribution and an onset in early childhood.

## 9. Type of Mutations in the Investigated Tumor Types

To assess the spectrum of somatic and germline *DICER1* mutations, we performed a systematic review of the literature. Our analysis yielded 246 cases of the four entities published in the literature (Table 1 and Appendix A). These included 15 ETMRs, 70 intracranial sarcomas, 43 pineoblastomas (PinB), and 118 pleuropulmonary blastomas (PPB). Although PPB is not localized in the CNS, we included it as an extracranial reference entity for DICER1 syndrome in pediatrics (Table 1).

We categorized *DICER1* mutations into missense, nonsense, and frameshift mutations, further distinguishing them as either germline or somatic (Table 1). In the case of ETMRs, 25 instances were identified, comprising 14 missense, three frameshift, and eight nonsense mutations, as derived from the published literature. Additionally, we identified one patient in the CNS-InterREST GPOH database with a missense mutation. In intracranial sarcomas, 70 cases revealed a total of 98 mutations, comprising 76 missense, 7 frameshift, and 15 nonsense mutations. Similarly, among 43 PinB cases, 41 mutations were identified, including seven missense, 17 nonsense, and 17 frameshift mutations. For our extracranial reference group (PPBs), we analyzed 118 cases, uncovering 145 mutations—65 missense, 42 nonsense, and 38 frameshifts. As anticipated, the determination of germline or somatic origin was not reported in all cases; consequently, not all 246 cases were represented in the figures.

Next, to assess the spectrum of mutations in these entities, we plotted them along the DICER1 protein sequence (Uniprot: Q9UPY3-1) and mapped them to the known functional domains (Figure 2). In ETMR, most somatic mutations accumulate in the RNase IIIb domain, while germline mutations more often affect the 5’ end of the gene. Almost all missense mutations occur in the RNase III domains, indicating a hotspot for somatic amino acid changes in this region. In contrast, most frameshift and nonsense mutations are dispersed over the rest of the *DICER1* gene. More than half of the mutations occurred in the RNase IIIb domain, leaving only 42% of all mutations outside this specific domain (Figure 3c).

Mutations in intracranial *DICER1* mutant sarcomas were also mainly localized to the RNase IIIb domain (81%, Figure 2 and Figure 3c). This might—at least—in part be explained by the fact that the cohort of Peruvian patients contributed to a high proportion of non-germline-associated *DICER1* mutant sarcomas, which are predominantly in the RNase IIIb domain [25]. The mutations are distributed fairly like those in ETMR, with most missense mutations affecting the RNase III domains. Unlike ETMR, sarcomas also tend to accumulate mutations in the DRBM, the RNase IIIa, and the Platform domains but do not show any mutational differences in the PACT and TRBP-binding domain and the DICER1 dsRNA-binding fold, which may indicate tissue-specific mutations or mechanisms. However, this difference could also be due to the low overall number of ETMR *DICER1* mutations. The frequency of germline mutations was only 19%, the lowest among all analyzed entities (Figure 3a). Additionally, sarcomas exhibited the highest proportion of missense mutations, with 78%.

Interestingly, the mutation distribution in pineoblastomas revealed a pattern distinct from those of the other analyzed entities. The mutations were more diffused across the entire gene, with 80% occurring outside the RNase IIIb domain (Figure 2 and Figure 3c). Importantly, we identified four missense mutations localized to the Helicase domain. Of the 42 mutations, only eight occurred in the RNase III domains. Apart from the differences in location, the distribution of mutation types is unlike that of ETMR and sarcomas. While missense mutations are the most common type in the other two entities, pineoblastomas show a distinct pattern, with frameshift and nonsense mutations being more prevalent than missense mutations. In pineoblastomas, missense mutations account for only 17% of cases, compared to approximately 40% in the other two entities (Figure 3b).

In our extracranial reference group (PPBs), the data distribution resembled that of ETMR and sarcoma mutations. Most somatic missense mutations accumulate in the RNase IIIb domain (43), with only three exceptions in the *DICER1* dsRNA-binding fold, in the PACT and TRBP-binding domain, and outside all domains. Frameshift and nonsense mutations are predominantly germline and tend to be spread more evenly across genes. Most germline mutations occur outside these domains (19). Importantly, Chi-squared analysis of the distribution of germline and somatic mutations occurring within or outside the RNase IIIb domain in the four entities showed significant enrichment of somatic mutations in RNase IIIb, specifically in PPB. However, this result should be interpreted with caution, as it may be influenced by the limited number of mutations available for the other entities (Figure 3d).

## 10. Conclusions

While the ratio of *DICER1*-mutant tumors is exceedingly low and has only recently been described, the mechanisms by which *DICER1* causes ETMR may be comparable to those of tumors that are C19 miRNA mutants. Lambo et al. proposed the formation of R-loops as an effector mechanism [16]. Most ETMR occur in patients < 3 years of age [15]. Thus, it is worth noting that in a small number of patients, this tumor, if survived, may be a harbinger of DICER1 syndrome. Although the time interval in these very young infants that may be open to permit early tumor detection is small, it is an important consideration in index families where the *DICER1* germline mutation may be known from birth. Similar considerations may be made in intracranial sarcoma *DICER1* mutants, where patients are typically considerably older than those with ETMR.

We would like to stress that the small sample sizes (also in the published cases) may not currently permit major, definitive conclusions. However, we believe that this investigation is particularly relevant, as it remains uncertain whether a genotype-phenotype correlation exists in these rare brain tumor types. Furthermore, it remains unclear whether the localization of mutations within specific functional domains may influence the predisposition to develop ETMR or DICER1-associated intracranial sarcomas.

*DICER1* is predominantly classified as a tumor suppressor gene due to the overall presence of loss-of-function mutations that often follow the “two-hit hypothesis,” as well as its critical role in maintaining genomic stability [3]. However, the occurrence of specific hotspot mutations in the RNase IIIb domain in some tumors, such as intracranial sarcomas or ETMR, raises the question of whether *DICER1* may function as an oncogene in these contexts. This hypothesis is further supported by the fact that these mutations may impair the ability to process specific miRNAs while retaining the capacity to process others, thus selectively deregulating miRNA pathways to promote oncogenesis. While these observations do not redefine *DICER1* as an oncogene, they underscore the need for further detailed molecular studies comparing the mechanisms of action of the mutations. Additionally, the presence of isolated cases with missense mutations outside the RNase domains suggests the potential involvement of a different mechanism in tumorigenesis. While a plethora of studies have focused on delineating the RNase IIIb domain, less is known about how mutations outside this domain could cause cancer [7,16,33].

In summary, our analysis collates the current knowledge on the genetic basis of ETMR and *DICER1* mutant intracranial sarcomas. We also delve into the various challenges inherent in treating patients with DICER1-associated tumors in the context of a germline mutation, such as the development of secondary tumors and the associated increase in morbidity. Thus, oncological multimodal therapy needs to be considered with caution, and predictive algorithms, as well as future studies that prognosticate the individual risk for the development of further neoplasias, are necessary. Such tools could inform clinical decision-making and contribute to reducing the considerable mortality rate in patients with this rare condition.

## Figures and Tables

**Figure 1 cancers-17-01513-f001:**
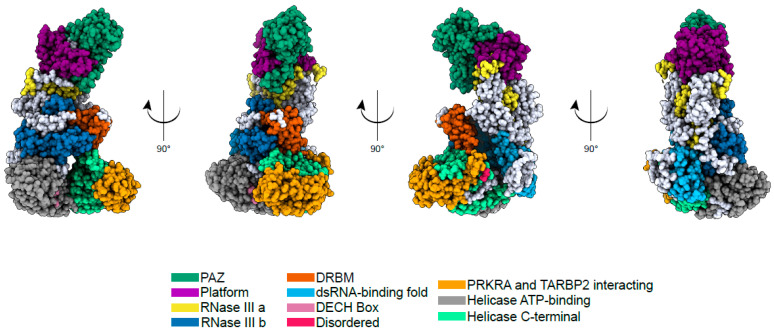
Cryo-EM structure of human DICER1. Cryo-EM structure of the human DICER1 protein generated by Lee et al. 2023 [4], visualized using ChimeraX.

**Figure 2 cancers-17-01513-f002:**
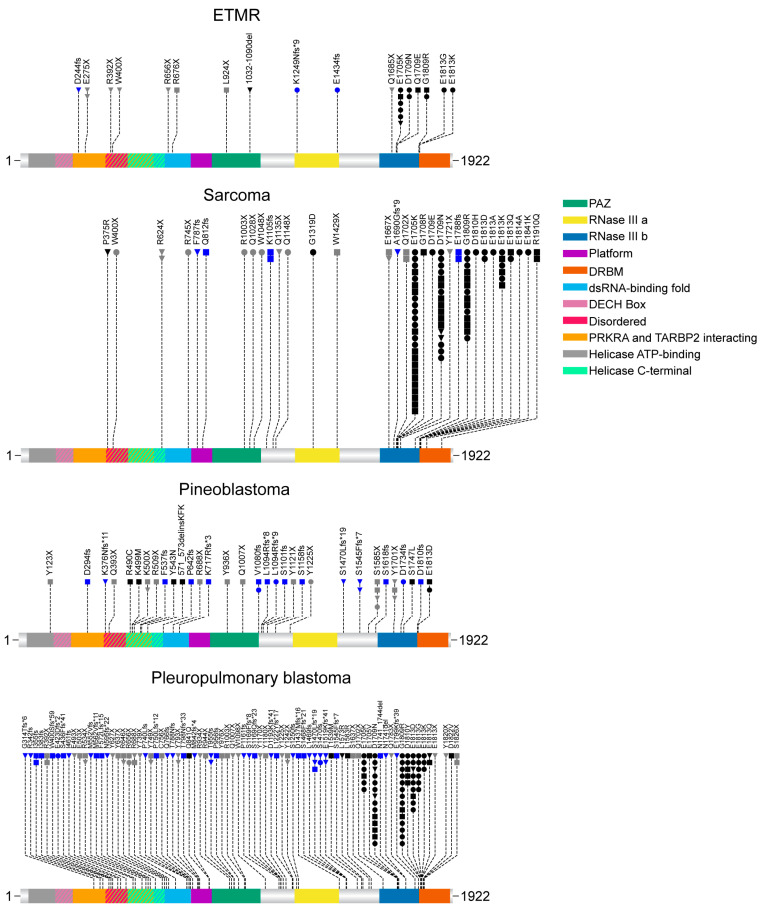
Spectrum of *DICER1* mutations in pediatric CNS tumors. Lollipop plots of all *DICER1* mutations included in this review, plotted along the DICER1 protein (Uniprot: Q9UPY3-1). Pleuropulmonary blastoma was included as a non-CNS, pediatric *DICER1*-mutant entity for comparison purposes. Germline mutations are represented as triangles, somatic mutations as circles, and N/As as squares. The black symbol color represents missense mutations, blue frameshift, and gray nonsense. Domains include Helicase ATP-binding (amino acids 51–227), DECH Box (175–178), PRKRA and TARBP2 interacting (256–595), Disordered (409–433), Helicase C-terminal (433–602), dsRNA-binding fold (630–722), Platform (752–895), PAZ (895–1042), RNase IIIa (1276–1403), RNase IIIb (1666–1824) and DRBM (1849–1914).

**Figure 3 cancers-17-01513-f003:**
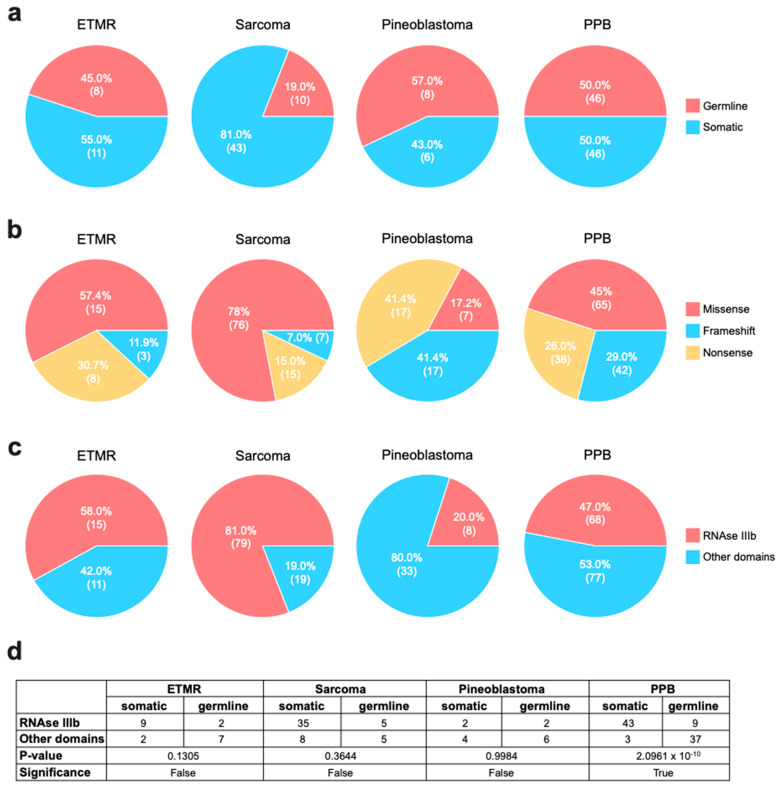
Distribution of *DICER1* mutations in pediatric CNS tumors. (**a**–**c**) Pie charts of mutation distributions within each of the studied entities, categorized as (**a**) germline or somatic, (**b**) missense, frameshift, or nonsense, and (**c**) affecting the RNase IIIb or any other domain. (**d**) Chi-square analysis results comparing the number of germline and somatic mutations occurring within the RNase IIIb domain vs. other domains of DICER1.

**Table 1 cancers-17-01513-t001:** Overview of the patients and mutations included in the analysis.

Variable	Overall	ETMR	Sarcoma	PinB	PPB
*Age*, *y*					
Median	6.8	3.7	11.9	8.3	3.7
Range	0.1–76	1–30	0.1–76	1–30	0.1–27
Gender					
Male	60	6	19	14	21
Female	79	8	26	8	47
Not Available	108	2	25	21	60
Total	247	16	70	43	118
*Mutation type*					
Missense	163	15	76		65
Nonsense	78	8	15	17	38
Frameshift	69	3	7	17	42
Total	310	26	98	41	145
*Germline/Somatic*					
Germline	73	9	10	8	46
Somatic	106	11	43	6	46
Not Available	131	6	45	27	53
Total	310	26	98	41	145

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
