# Peer review of "DICER1 Mutational Spectrum in Intracranial CNS-Neoplasias—A Review and a Report from the CNS-InterREST GPOH Study Center"

_cancers, 2025, doi:10.3390/cancers17091513_

Round 1
Reviewer 1 Report
Comments and Suggestions for Authors
The genetic landscape of DICER1 variants in different pediatric brain tumors is investigated in the review by Manea et al. The paper integrates literature review and cohort analysis to describe mutational patterns, clinical implications, provide useful clinical and molecular insights, and advocate implementing ad hoc surveillance for DICER1 syndrome cases.
The review focuses on specific tumor types and provides evidence that, in addition to those historically associated with DICER1 syndrome, ETMR may also be underpinned by mechanisms involving DICER1, suggesting its role as an oncogene and not only as a tumor suppressor.
A total of 248 cases (ETMR, sarcoma, pineoblastoma, PPB) from the literature were included in the analyzed cohort.
The sample size of ETMR with a hypothetical role of dysregulation of some processes regulated by DICER1 (e.g. in miRNA genesis, mechanistic differences or overlap between C19MC-amplified and DICER1-mutant ETMR) is very small. The link between the observed alterations in DICER1 and a direct involvement in ETMR tumorigenesis requires further mechanistic studies.
The results reiterate the need for genotype-phenotype studies (the role of RNase IIIb domain variants) to optimize surveillance protocols, integrating intracranial imaging in DICER1 surveillance, potentially improving the outcome of patients affected by aggressive tumors at a pediatric age. However, stronger validation is needed.
In spite of this, the limitations are explained and the study is worthy of publication in its current form.
Reviewer 2 Report
Comments and Suggestions for Authors
In this narrative review the authors summarized the evidence on DICER1 mutations by combining bibliographic and genetic data. The genetic evidence derived from a German database. A total of 246 cases reported of different neoplasias were discussed. The DICER1 molecular structure, the epidemiology of the DICER1 syndrome, and DICER1 mutations associated with cancer types, are the focus of this report.
Concerns
1. The frequency of DICER1 mutations in the neoplasias examined is not reported. There are reported cases of DICER1 mutations, but it has not be reported how often such mutations are observed in these neoplasias.
2. The https://doi.org/10.1007/s11060-022-03994-w could be added.
3. Partially the conclusions are not consistent. The reason is that we don't know in what settings these mutations appear to suggest actions such as the "...we propose preforming cerebral MRI monitoring once per year in addition...", line 294
Reviewer 3 Report
Comments and Suggestions for Authors
This is a comprehensive discussion of the genetics and potential underlying mechanisms of pathogenesis for DICER1-related CNS tumors. While I appreciate the summary of data and hypotheses presented, the surveillance recommendations provided are not sufficiently supported by the data presented. Recommendations for surveillance should consider prevalence within the risk populations, age range of risk, impact of early diagnosis on morbidity and mortality and also potential risks of the proposed modality. These discussions, while worthy of ongoing discussion, seem outside the scope of this manuscript.
Reviewer 4 Report
Comments and Suggestions for Authors
This manuscript compiles and synthesizes the mutational spectrum of DICER1 in intracranial CNS neoplasms in addition to neural findings from the CNS-InterREST GPOH study center. While the subject addresses the interest of the neurologic community, there are a few issues I would like to point out that should be fixed.
1.The introduction provides a general overview of DICER1 syndrome, however, a more specific description of the biological impact DICER 1 mutations cause in neural tissues would help strengthen the paper.
2.The region about mutations and the associated cancer types is detailed but not explicitly focused on neural tumors. Discussing peripheral neural tumors such as pleuropulmonary blastoma non neural tumors undermines the intent of the paper.
3. The references in the text do not follow citation of MDPI format.
4. The study's reliance on a relatively small sample size from the CNS-InterREST database is a limitation.
5. The literature review component of the manuscript appears to be selective and incomplete. The authors claim to have analyzed 248 cases from published literature, yet the discussion of these cases is fragmented and lacks a cohesive synthesis.
6. There was no clear indication and labeling in figure 2 which was used to map DICER I mutations along the sequence of the protein and as such, the importance of the presented information is lost.
7. The conclusions are overly optimistic. The authors should temper their statements regarding the potential clinical implications of their findings and emphasize the need for further research.
Round 2
Reviewer 3 Report
Comments and Suggestions for Authors
The authors have edited some points to reflect the suggestions of the other reviewers but have not adequately addressed concerns related to their recommendation for annual brain MRI for all individuals with germline DICER1. I think it is important to note that the points related to prevalence/numerical risks of disease, potential burdens and risks of surveillance (including need for sedation in young children), age range of surveillance, potential impact of early detection on outcome (less certain for CNS than for other tumors), risk for false positive findings and other important points related to whether universal surveillance for individuals with germline DICER1 should all undergo annual MRI (including how long this should continue) remains unaddressed. Importantly, frequency data from somatically mutated tumors without germline data cannot be used to support surveillance in individuals with germline variants.
Overall, the authors must ethically consider the potential ramifications of such a striking recommendation for ongoing annual brain MRI on the >>99% of individuals with germline DICER1 who do not develop a brain tumor but yet represent 1 in every 8000 individuals worldwide. To balance this with early detection for a small and yet clinically important group of very rare patients who do develop a brain tumor is important and not yet addressed here.
Author Response
Comment:
The authors have edited some points to reflect the suggestions of the other reviewers but have not adequately addressed concerns related to their recommendation for annual brain MRI for all individuals with germline DICER1. I think it is important to note that the points related to prevalence/numerical risks of disease, potential burdens and risks of surveillance (including need for sedation in young children), age range of surveillance, potential impact of early detection on outcome (less certain for CNS than for other tumors), risk for false positive findings and other important points related to whether universal surveillance for individuals with germline DICER1 should all undergo annual MRI (including how long this should continue) remains unaddressed. Importantly, frequency data from somatically mutated tumors without germline data cannot be used to support surveillance in individuals with germline variants.
Overall, the authors must ethically consider the potential ramifications of such a striking recommendation for ongoing annual brain MRI on the >>99% of individuals with germline DICER1 who do not develop a brain tumor but yet represent 1 in every 8000 individuals worldwide. To balance this with early detection for a small and yet clinically important group of very rare patients who do develop a brain tumor is important and not yet addressed here.
Authors' response:
We thank the Reviewer for this valuable feedback. After careful consideration of the burdens and risks listed by the Reviewer, we have decided to remove the surveillance suggestion from our review and have edited the relevant sections of the review accordingly.
Reviewer 4 Report
Comments and Suggestions for Authors
The author has provided a good response to the questions raised, and it is recommended for publication.
Author Response
Comment:
The author has provided a good response to the questions raised, and it is recommended for publication.
Authors' reply:
We thank Reviewer 4 for their feedback.
Round 3
Reviewer 3 Report
Comments and Suggestions for Authors
Thank you to the authors for this excellent paper which is a great contribution to the literature and for their willingness to address the concerns raised.